# Tillage type and sentinel insect species affect the relative prevalence of the entomopathogenic fungus, *Metarhizium robertsii*, in soil

**Shea A. W. Tillotson[1], Christina A. Voortman[1], John M. Wallace[2], Mary E. Barbercheck●[1]***

**1** Department of Entomology, The Pennsylvania State University, University Park, PA, United States of America, **2** Department of Plant Science, The Pennsylvania State University, University Park, PA, United States of America

\* meb34@psu.edu

**Data Availability Statement:** The data underlying the results presented are available from OSF database (DOI: 10.17605/OSF.IO/XUGQZ; https://osf.io/xugqz/).

## Abstract

Because the use of synthetic agrochemicals is generally not allowed in organic crop production systems, growers rely on natural substances and processes, such as microbial control, to suppress insect pests. Reduced tillage practices are associated with beneficial soil organisms, such as entomopathogenic fungi, that can contribute to the natural control of insect pests. The impacts of management, such as tillage, in a cropping system can affect soil biota in the current season and can also persist over time as legacy effects. We investigated the in-season and legacy effects of soil management in four, three-year organic feed grain and forage production systems that varied in number and intensity of soil disturbances on the relative prevalence of the entomopathogenic fungus, *Metarhizium robertsii*. Employing sentinel bait assays with *Tenebrio molitor* and *Galleria mellonella*, we found that relative prevalence, measured as infection rate of sentinel insects, was lowest in systems utilizing a shallow high-speed disk (*G. mellonella*: 14%; *T. molitor*: 23%) in the current and previous seasons compared to systems that included inversion and non-inversion tillage (*G. mellonella*: 22%; *T. molitor*: 34%) or no-till planting (*G. mellonella*: 21%; *T. molitor*: 30%,). There was no difference in prevalence in systems that included the use of a high-speed disk compared to a perennial hay crop (*G. mellonella*: 16%; *T. molitor*: 28%). There were no negative legacy effects of inversion tillage on the prevalence of *M. robertsii* in subsequent crops. Sentinel assays with *G. mellonella* (19%) produced overall lower estimates of relative prevalence of *M. robertsii* than *T. molitor* (29%) but the association of relative prevalence with environmental variables was greater in assays with *G. mellonella*. We suggest that the use of occasional inversion tillage is not damaging to populations of *M. robertsii* in soil and that surveys using assays with multiple sentinel insect species will improve our ability to understand the effects of agricultural practices on entomopathogenic fungi.

**Funding:** Research was partially funded by USDA National Institute of Food and Agriculture, Organic Research and Education Initiative (OREI), awarded to J.M.W. and M.E.B., under award number 2020-51300-32378, The funders had no role in the study design, data collection and analysis, decision to publish, or preparation of the manuscript.

**Competing interests:** The authors have declared that no competing interests exist.

## Introduction

Soil health, the ability of soil to sustain life while also supporting a complex ecosystem, is foundational to organic agriculture [1–3] and biodiversity is a vital component of soil health in agroecosystems [4,5]. Biodiversity tends to be greater where organic practices are utilized compared to agroecosystems where conventional practices are used, especially where synthetic agrochemicals and fertilizers are applied [6,7]. Soil health integrates chemical, physical, and biological characteristics of soil, and many indicators to quantify soil health have been developed [8–10]. Management practices that can positively contribute to soil health in both organic and conventional systems include maintenance of plant residue cover on the soil surface, living roots, spatial and temporal diversity within a cropping system, and reduced soil disturbance [11–14]. Reduced tillage practices are associated with conservation of soil microbial communities that include entomopathogenic fungi, which contribute to the ecosystem service of biological control of insects [15–17].

Fungi are the most common of insect pathogens in soil [15,18,19]. Encouraging establishment and conserving entomopathogenic fungi (EPF) can contribute to the suppression of insect pests due to their ability to infect and kill insects directly, or for some of the most common soilborne hypocrealean EPF, indirectly by endophytically colonizing host plants [20–23]. The EPF, *Metarhizium robertsii* Bisch, Rehner & Humber (Hypocreales: Clavicipitaceae), commonly found in agricultural soil, causes direct mortality by infecting and causing disease in susceptible insects via contact with conidia. It can also impact arthropods indirectly as a beneficial facultative endophyte of a broad range of plant species, where it can suppress insect growth and promote plant growth [15,23–25].

While many non-organic growers in the Northeastern US have adopted conservation or reduced tillage practices, organic growers tend to rely on inversion and other forms of tillage to manage weeds and crop residues, incorporate fertility amendments, and prepare seedbeds [26–30]. Soil disturbance caused by frequent or intense tillage can suppress soil biological communities on the soil surface and within the soil and affect the functioning of agroecosystems [17,31]. Tillage can disturb fungal communities and reduce the prevalence of *Metarhizium* spp. compared to soil with minimal disturbance [16,17,32]; however, *M. robertsii* has also been detected in organic production systems even where frequent and/or intensive soil disturbance occurs [32–34].

Organic growers that are interested in building and maintaining soil health and its associated benefits may consider reducing the level of disturbance that they impose on their soil by reducing tillage in a variety of ways. Cover crop-based, organic rotational no-till management practices can improve soil conservation in organic feed and forage systems through no-till planting cash crops into a roller-crimped mat of cover crop residue [35–37]. Including a perennial crop, e.g., alfalfa (*Medicago sativa* L.), into a crop rotation can reduce disturbance over time by removing tillage completely after establishment [38]. Some growers use chisel plowing, a non-inversion approach, to manage soil and residues [39], which can cause less disturbance to soil than inversion tillage with a moldboard plow. Another option for growers who are intent on reducing soil disturbance intensity is the shallow tillage high-speed disk (HSD) [40]. This tillage tool mixes soil to a shallow depth but does not invert the soil column. The HSD penetrates the soil to a depth up to 10 cm, which is 5–10 cm shallower than typical for the moldboard plow, potentially reducing disturbance to biological communities that are deeper in the soil profile. This tool may balance the tradeoff between maintaining soil health by reducing tillage depth without sacrificing the ability to incorporate cover crop residues and fertility amendments, manage weeds, and prepare seedbeds for crops. However, little is understood about the relative effects of the HSD as a primary tillage tool on the prevalence of

soilborne EPF, especially when coupled with other conservation practices such as the use of cover crops, crop rotation, and organic amendments.

Here we report the results of research to determine the effects of number and intensity of soil management events and associated abiotic soil properties on the relative prevalence of *M. robertsii* in four, three-year annual feed grain and perennial forage systems managed to vary in soil disturbance. We investigated both in-season and legacy [i.e., carry over] effects of management using two sentinel bait insect species, *Galleria mellonella* L. and *Tenebrio molitor* L. in bioassays to test the relative efficiency in detecting differences in relative prevalence among the experimental systems [16,41]. We hypothesized that cropping systems with 1) the greatest frequency and intensity of disturbance would have the lowest relative prevalence of *M. robertsii* in the soil; 2) systems with an intermediate frequency and intensity of disturbance would have an intermediate relative prevalence of *M. robertsii* in the soil. Further, we hypothesized that 3) the efficacy of sentinel insect species used in bioassays to detect *M. robertsii* would vary with the level of disturbance and abiotic conditions.

## Methods

### Site description

This experiment was conducted at the Russell E. Larson Agricultural Research Center in Pennsylvania Furnace, PA (40.723165, -77.929840). This experiment is approximately 4 hectares in area, has been managed according to USDA organic regulations since 2011, and received organic certification in 2014 [42]. In 2023, this site was in Zone 6b of the USDA Plant Hardiness Zones [43]. Annually, this site experiences an average of 1,000 mm of precipitation. Average annual temperatures range from 5˚C to 28˚C. Soils at the site are representative of the Hagerstown Soil Series according to the USDA Natural Resources Conservation Service soil classification system and includes mostly silt loam [44].

### Experimental design

The field experiment was comprised of four organic cropping systems that differed in cover crop species, establishment and terminations method; tillage tool, timing, and frequency of tillage; and in-season crop management. Winter cover crops were grown between each annual cash crop. Systems 1 through 3 consisted of a three-year, annual crop rotation of corn (*Zea mays* L., Viking O.45-88-P)—soybean (*Glycine max* L. Merr., Viking O.2155 N)—wheat (*Triticum aestivum* L., Malabar) in which all crops in the rotation were present in each year (i.e., full entry design), and a fourth system comprised of a mixture of alfalfa (*Medicago sativa* L., King's 544 PLH) and orchardgrass (*Dactylis glomerata* L., King's Echelon). To maintain the crop rotation in the full entry experiment, each crop entry was initiated with a different crop in the sequence in 2021. For ease, where crop entry is discussed, the abbreviations C-S-W, S-W-C, or W-C-S will be used to indicate the crop sequence, where C denotes corn, S denotes soybean, and W denotes wheat, listed in order of year in which the crop was grown in 2021, 2022, and 2023. The experiment was implemented in a randomized complete block design. Each treatment was replicated four times totaling 48 experimental plots, each measuring 6.1m by 48.7m (S Tables 1 and 2 in S1 File).

The four systems, or treatments, were designed as follows. System 1 used full inversion tillage before corn and soybean similarly to commercial organic grain farms in the Northeast US, while also integrating cover crops using reduced tillage methods. Winter wheat was sown in October following the use of a chisel plow to a depth of 15 cm at 202 kg ha[-1] (S Tables 2 and 3 in S1 File). In early March, medium red clover (*Trifolium pratense* L, Albert Lea Seed, VNS) was frost-seeded at a rate of 17 kg ha[-1] into winter wheat using a no-till grain drill. After wheat

was harvested in July, the medium red clover was allowed to grow and then mowed to a height of 5 cm in early October. Cereal rye (*Secale cereale* L., Aroostook) was then no-till drilled at a seeding rate of 33 kg ha$^{-1}$ in mid-October into established red clover. In late May to early June, corn was planted at 86,000 seed ha$^{-1}$ into a seedbed created using a moldboard plow set to a depth of 20 cm. A cover crop mixture of annual ryegrass (*Lolium perenne* spp. *multiflorum*, Kodiak), forage radish (*Raphanus sativus*, Organic Tapmaster), and crimson clover (*Trifolium incarnatum*, Dixie) was interseeded into standing corn (V4 vegetative growth stage) after last cultivation at 28 kg ha$^{-1}$ using a high-clearance, no-till grain drill. The interseeded cover crop mixture was allowed to growth through the following fall and spring, and in late May soybean was planted 590,000 seed ha$^{-1}$ into a seedbed created using a moldboard plow set to a depth of 20 cm.

System 2 used shallow tillage with a compact high-speed disk (HSD), intended to reduce the intensity and depth of soil disturbance before each cash crop and to understand the effects of the HSD on multiple soil health and agronomic performance indicators. The HSD uses shallow non-inversion tillage that mixes the soil to a depth of approximately 10 cm. The number of HSD passes necessary to achieve an adequate seedbed differed by crop and year, depending on soil and crop residue conditions. The HSD was used before all cash crop plantings in System 2. Wheat was planted in late October at a seeding rate of 202 kg ha$^{-1}$. After wheat harvest in July, a cover crop mixture of oat (*Avena sativa* L., Jerry), forage radish, and Austrian winter pea (*Pisum sativum* L., Albert Lea Seed, VNS) was drill-seeded at a seeding rate of 22 kg ha$^{-1}$, 1 kg ha$^{-1}$, 32 kg ha$^{-1}$, respectively. Corn was planted in late May or early June, at a rate of 86,000 seed ha$^{-1}$. Soybeans were planted in late May or early June following high-speed disk tillage at a rate of 590,000 seed ha$^{-1}$. After corn harvest, cereal rye was seeded at a rate of 67 kg ha$^{-1}$.

System 3 was a reduced tillage system intended to reduce disturbance to the extent possible in an organic grain rotation and used no-till planting in soybean and relay cover cropping practices in wheat. System 3 had the lowest tillage intensity among the annual cropping systems, and the longest period between primary tillage events. A moldboard plow was used before planting corn at a seeding rate of 86,000 seed ha$^{-1}$. This was the only inversion tillage event in the 3-yr crop rotation. Cereal rye was drill seeded at a seeding rate of 135 kg ha$^{-1}$ following harvest of corn. The cereal rye was terminated using a roller-crimper in the following spring before no-till planting soybean at a rate of 590,000 seed ha$^{-1}$ into the mat of roll-crimped cereal rye. In October, wheat was drilled at a seeding rate of 202 kg ha$^{-1}$ following a single pass with the HSD set at 5–10 cm depth. In March, medium red clover was frost drill-seeded into wheat at a rate of 16 kg ha$^{1}$.

System 4 used perennial alfalfa-orchardgrass as a minimal soil disturbance baseline to compare to the three annual cropping systems. In System 4, the alfalfa (29.1 kg ha$^{-1}$)–orchardgrass (8.9 kg ha$^{-1}$) mixture followed wheat harvest and chisel plowing and disking. Because the experiment was full entry, and alfalfa followed wheat in the rotation, the number of alfalfa plots differed by year. In the crop rotation entry that started with corn in 2021, there were four alfalfa plots. In 2022, an additional four plots were planted to alfalfa, and in 2023, four more plots were planted to alfalfa to total 12 alfalfa plots (S Tables 2 and 3 in S1 File).

## Soil disturbance

We used the number of field operations and a soil disturbance rating (SDR), from the USDA Natural Resources Conservation Service, to represent the frequency and intensity of soil disturbance from machinery for each system [45]. The six components of the SDR include soil inversion, soil mixing, soil compaction, soil shattering, soil lifting, and soil aeration. Each component of the SDR is assigned an intensity rating that ranges from 0 (least intense) to 5 (most

**Table 1. Annual and accumulated number of disturbances and soil disturbance ratings (SDR) by crop entry, year, crop and experimental system.**

| Entry | Year | Crop | Annual Number of Disturbances | | | | Annual SDR | | | | Rotation Number of Disturbances | | | | Rotation SDR | | | |
|---|---|---|---|---|---|---|---|---|---|---|---|---|---|---|---|---|---|---|
| | | | Sys 1 | Sys 2 | Sys 3 | Sys 4 | Sys 1 | Sys 2 | Sys 3 | Sys 4 | Sys 1 | Sys 2 | Sys 3 | Sys 4 | Sys 1 | Sys 2 | Sys 3 | Sys 4 |
| 1 | 2021 | Soy | 21 | 21 | 13 | 21 | 304 | 310 | 160 | 304 | 38 | 39 | 32 | 37 | 525 | 574 | 429 | 520 |
| | 2022 | Wheat | 7 | 6 | 7 | 9 | 38 | 74 | 38 | 117 | 45 | 45 | 39 | 46 | 563 | 648 | 467 | 637 |
| | 2023 | Corn | 16 | 21 | 19 | - | 240 | 330 | 291 | - | 61 | 66 | 58 | - | 803 | 978 | 758 | - |
| | | Alfalfa | - | - | - | 15 | - | - | - | 124 | - | - | - | 61 | - | - | - | 761 |
| Entry 1 Rotation Disturbance | | | 44 | 48 | 39 | 45 | 582 | 714 | 489 | 545 | 61 | 66 | 58 | 61 | 803 | 978 | 758 | 761 |
| 2 | 2021 | Wheat | 4 | 6 | 4 | 9 | 29 | 74 | 29 | 107 | 25 | 26 | 24 | 30 | 303 | 340 | 294 | 382 |
| | 2022 | Corn | 15 | 15 | 16 | - | 205 | 226 | 229 | - | 19 | 21 | 20 | - | 508 | 566 | 523 | - |
| | | Alfalfa | - | - | - | 13 | - | - | - | 93 | - | - | - | 22 | - | - | - | 475 |
| | 2023 | Soy | 19 | 18 | 7 | - | 311 | 318 | 83 | - | 38 | 40 | 26 | - | 819 | 884 | 606 | - |
| | | Alfalfa | - | - | - | 12 | - | - | - | 36 | - | - | - | 33 | - | - | - | 511 |
| Entry 2 Rotation Disturbance | | | 38 | 39 | 27 | 21–22 | 545 | 618 | 341 | 143–200 | 38 | 40 | 26 | 52–63 | 819 | 884 | 606 | 382–511 |
| 3 | 2021 | Corn | 19 | 21 | 21 | - | 231 | 274 | 279 | - | 25 | 29 | 27 | - | 272 | 380 | 320 | - |
| | | Alfalfa | - | - | - | 14 | - | - | - | 126 | - | - | - | 21 | - | - | - | 232 |
| | 2022 | Soy | 20 | 19 | 10 | - | 292 | 279 | 123 | - | 45 | 48 | 37 | - | 564 | 654 | 443 | - |
| | | Alfalfa | - | - | - | 13 | - | - | - | 39 | - | - | - | 34 | - | - | - | 271 |
| | 2023 | Wheat | 6 | 6 | 6 | - | 31 | 88 | 31 | - | 51 | 54 | 43 | - | 595 | 742 | 474 | - |
| | | Alfalfa | - | - | - | 13 | - | - | - | 39 | - | - | - | 47 | - | - | - | 310 |
| Entry 3 Rotation Disturbance | | | 45 | 46 | 37 | 13–14 | 554 | 641 | 433 | 39–126 | 51 | 54 | 43 | 21–47 | 595 | 742 | 474 | 232–310 |

intense) for each field operation. The total rating for an operation is the sum of the six components. Therefore, SDR ranges from 0, which is the least intense level of disturbance from a field operation, to 30, which is the most intense level of disturbance. The tillage implement with the highest SDR was the moldboard plow, with a rating of 29. The high-speed disk was assigned a rating of 24. No-till planting had an associated SDR value of 5.

The number of operations and SDR values for each system were summed starting on January 1 of each year and accumulated through December 31 to provide an annual rating (Table 1). Annual number of operations and SDR were accumulated across each year from 2021 through 2023 to provide estimates of disturbance across the crop sequence [rotation] in a system. To determine the effects of in-season and rotation number and intensity on *M. robertsii* during a year, we accumulated the numbers of disturbances and SDR that occurred before each soil sample from January 1 of the sample year and from January 1, 2021, respectively. We included number of days from the most recent field operation as a factor in analyses to determine the effects of the time elapsed since disturbance because we considered that recency of disturbance could affect observed relative prevalence.

Multiple consecutive passes using the same equipment was common during the experiment. For example, in System 2 multiple passes with the high-speed disk were sometimes needed to sufficiently incorporate cover crop residue. These multiple passes can account for the greater annual and rotation numbers of disturbances and SDR in System 2 than in System 1 (inversion), even though the HSD has a lower SDR than the moldboard plow used in System 1. System 2 was designed to represent an intermediate level of disturbance. Even though management in System 2 sometimes resulted in the greatest overall SDR, we still considered it to represent an intermediate level of disturbance due to the shallow non-inversion action of the

HSD. The multiple consecutive uses of the same implement within the same field operation may or may not have similar effects on biological communities as a single use [46]. Therefore, while the number of disturbances and their associated SDR are useful measures for quantifying and comparing the relative levels of disturbance across systems, we acknowledge that there are challenges to its use and interpretation that remain to be addressed through research.

## Relative prevalence of *M. robertsii*

To assess the effects of system on the relative prevalence of *M. robertsii*, we conducted a standard soil bioassay using larval insects as a sentinel bait [47]. We collected pre-plant (April) and pre-harvest (September) composite soil samples comprised of 11 cores with a soil probe (2.5 cm x 20 cm) from all treatment plots. We collected post-plant (June) samples from plots in the corn and soybean phases of the rotation in Systems 1–3 in 2021, 2022, and 2023 (S Table 3 in S1 File). In the laboratory, we homogenized the composite sample from each treatment plot by crumbling soil clumps and removing plant residue and rocks. From each sample, we removed two, 250 ml subsamples for soil assays to detect entomopathogenic fungi (EPF) and one 250 ml subsample for soil fertility analysis.

We transferred the two, 250 ml subsamples for the entomopathogen assays to two 500 ml plastic deli containers (12 cm x 8.5 cm). Previous studies have found that results of sentinel bait assays to detect entomopathogenic fungi could vary by insect used [16]. Therefore, we conducted the assays concurrently and separately using last instar *T. molitor* L. (Coleoptera: Tenebrionidae) and last instar *G. mellonella* L. (Lepidoptera:Pyralidae). Fifteen last instar *G. mellonella* or 15 last instar *T. molitor* were added to each assay arena. We stored the prepared assay arenas at room temperature in the dark for ten days, after which, we retrieved the larvae from the soil for assessment. We classified the larvae as alive, dead from causes other than EPF, or potentially infected with *M. robertsii* and other EPF based upon morphological characteristics [48]. We recorded and then discarded larvae that were alive or dead from other causes. We rinsed potentially infected larvae with tap water to remove soil particles, then rinsed with 80% ethanol and then tap water. We then placed the larvae in humid chambers consisting of plastic Solo® condiment cups (7 cm x 3 cm) with a small piece of Whatman No. 1 filter paper to maintain humidity and allow for sporulation. After 7–10 days at room temperature, we initially determined infection by *M. robertsii* by morphology and the presence of characteristic green spores on the surface of the insect cadavers. We confirmed the identity of fungi initially identified as *M. robertsii* by sequencing the translation elongation factor-1 alpha (TEF-1α) of a random subset of isolates from infected *G. mellonella* and *T. molitor* from each System by the methods of Kepler et al. [49] (S Table 4 in S1 File). We observed infection by *Beauveria* Vuill. (Hypocreales: Cordycipitaceae) in only four assay insects among the 15,120 insects used in assays during the experiment. Therefore, we excluded *Beauveria* from further consideration. We did not detect any other EPF with the assay method used.

## Soil properties

We submitted a 250 ml subsample of soil from each plot to the Agricultural Analytical Services Laboratory of The Pennsylvania State University (University Park, PA) for analysis of the following characteristics: proportions of sand, silt, and clay; pH, electrical conductivity, salts, phosphorus (P), potassium (K), magnesium (Mg), calcium (Ca), cation exchange capacity (CEC), soil organic matter by loss-on-ignition (SOM-LOI), and the trace elements zinc (Zn), copper (Cu), and sulfur (S). We used 5 g of each soil sample to determine the concentration of permanganate oxidizable carbon (POX-C) concentrations as an indicator of labile, biologically

active soil carbon [50]. We used a 500 ml subsample to determine gravimetric moisture [51] and soil matric potential [52].

On two dates (4 October 2022, 11 October 2023), we measured soil compaction in each plot using a digital recording penetrometer (CYNST, Anhui East Electronic Technology Co., Ltd.) to a maximum depth of 45.7 cm. We measured compaction in three random locations in each plot and averaged the values to produce mean compaction value for the plot. Penetrometer data included: compaction in mPa in 2.54 cm increments to a maximum depth of 45.7 cm, maximum depth of penetrometer readings, and depth of maximum compaction.

## Statistical analysis

In univariate analyses, we used generalized linear mixed-effect models to analyze relative prevalence of *M. robertsii*, expressed as the proportion of *T. molitor* and *G. mellonella* on which the fungus was detected [53]. We compared the null random-effects only model to fully fitted models containing main effects and interactions by adding one effect at a time. The function "anova" [54] was used to determine significance of fixed main effects and interactions using log-likelihood ratio tests and the Wald $\chi^2$ test statistic. To meet assumptions of normality, equality of variances, and to reduce heterogeneity of variances, we utilized distribution options in R 4.2.2 [54]. We transformed percentages and proportions using square root arcsine transformation. The generalized linear mixed effects models were created with the function "glmmTMB" in the package *glmmTMB* [55] or used binomial distributions with the function "weights" set to the total number of *G. mellonella* or *T. molitor* used in each plot to account for the proportion of sentinel insects infected.

When models were significant, we used the package *emmeans* [56], to compare system effects with Tukey HSD *post hoc* tests. To determine mean prevalence for each individual grouping [by cash crop, system], we used the function "group_by" in the package *dplyr* [57]. Where the model was significant, we conducted pairwise comparisons with a Tukey HSD post hoc test in the package *emmeans* [56]. We considered the results of analyses significant at p < 0.05. Untransformed data are presented in the tables and figures.

We conducted data analyses to answer the following questions:

1. Did the relative prevalence of *M. robertsii* differ among systems?
   The relative prevalence based on the proportions of *T. molitor* and *G. mellonella* infected by *M. robertsii* in bioassays of soil were modeled separately using a generalized linear mixed effects model. System, entry point (C-S-W, S-W-C, W-C-S), year, and interactions between system and entry point, system and year, and entry point and year, and the three-way interaction between, system, entry, and year were used as fixed effects. To account for non-independence, random effects included assay date nested within plot, within entry, within block. This accounted for repeated measures of the same plot three times in each year over three years and block effects. We report statistical values only for significant effects in the results.

2. Did the relative prevalence of *M. robertsii* based on sentinel *T. molitor* and *G. mellonella* differ?
   We compared the proportions of *M. robertsii*-infected *T. molitor* and *G. mellonella* using a generalized linear mixed effects model. The proportion of infected *T. molitor* was set as the response variable and proportion of infected *G. mellonella* was set as a predictor variable. Year and system were set as fixed effects. Random effects included assay date nested within year, within crop, within block to account for repeatedly measuring the same plot three times per year and to account for non-independence.

3. Did the legacy of differences in tillage used in soybeans in 2021 affect the relative prevalence of *M. robertsii* in corn in 2023?

The main tillage treatments were implemented in the soybean phase of the experiment while management of the wheat and corn phases were similar among the systems (S Table 3 in S1 File). To determine the presence of a legacy effect due to management differences in soybean on the relative prevalence of *M. robertsii* in corn, we used the function "glmmTMB" in the package *glmmTMB* [55] with data from sentinel assays of soil from soybean plots in 2021 and corn plots in 2023. We ran two generalized linear mixed effects models. Both models included assay date nested within block as the random effect to account for non-independence and repeated sampling, and system as the fixed effect. Only data from 2023 was analyzed to determine potential legacy effects from previous management in the system for the complete 3-year crop sequence. When the effects of system were significant for the relative prevalence of *M. robertsii*, we used a Tukey pairwise *post hoc* test to compare the legacy effect of system with the function "emmeans" from the package *emmeans* [56].

4. Did the environmental conditions associated with each system affect the prevalence of *M. robertsii*?

To understand the relationship between disturbance and environmental variables associated with relative prevalence of *M. robertsii* in each system, we used forward selection stepwise multiple regression in JMP®, Version 17 (SAS Institute Inc., Cary, NC, USA). The initial pool of disturbance and soil variables included: the day of year that samples were collected and number of days since a field operation (disturbance); the number of disturbances and SDR accumulated from January 1 of the sample year until the date of soil sample (annual disturbance); the number of field operations and SDR accumulated from the beginning of the experiment until the date of soil sample (rotation disturbance); soil moisture (gravimetric soil moisture and matric potential); soil pH; electrical conductivity and salt content; cation exchange capacity; percent organic matter by loss-on-ignition and POX-C; P, K, Mg, Ca, zinc, copper and sulfur content; and percent sand, silt and clay. A similar multivariate analysis was conducted for the dates on which soil compaction data was available, limiting the measures of the relative prevalence of *M. robertsii*, disturbance, and soil properties to the dates that corresponded most closely to those on which compaction was measured. The final models were those with the lowest Akaike information criterion. We tested for multicollinearity of variables using pairwise correlations among the disturbance and soil variables. When a significant correlation between variables was detected, the variable with the greatest contribution (greatest F value, lowest p value) to variation in percentage of insects infected by *M. robertsii* was retained for use in further analyses to understand the combined effects of disturbance and environmental factors on the relative prevalence of *M. robertsii*.

To visualize the relationships between the systems, crop and environmental factors we used principal components analysis (PCA) in CANOCO for Windows version 5.0 [59]. The initial pool of disturbance and soil variables included: the number of disturbances and SDR accumulated from January 1 to December 31 of the sample year [annual disturbance]; the number of field operations and SDR accumulated from the beginning of the experiment until December 31 of the sample year (rotation disturbance); mean soil moisture (matric potential); soil pH; electrical conductivity and salt content; cation exchange capacity; percent organic matter by loss-on-ignition and POX-C; P, K, Mg, Ca, zinc, copper and sulfur content; and percent sand, silt and clay. Only environmental variables with at least 20% fit to the ordination space are visualized on biplots [59].

# Results

The number and SDR varied by system and crop sequence (Table 1). The numbers and intensities (SDR) of disturbance among systems, in order from greatest to least, were System 2, 1, 3, and 4.

## Relative prevalence of *M. robertsii*

Based on morphological and molecular methods, all fungal isolates from the sentinel insect assays were confirmed as *M. robertsii* [49]. These results are consistent with extensive sampling of soil from this site over the last 10 years [24,25,34].

1) Did relative prevalence of *M. robertsii* differ among systems?

Based on sentinel assays with *G. mellonella*, system had a significant effect on the relative prevalence of *M. robertsii* ($\chi^2$ = 12.61, df = 3, p = 0.0056). Relative prevalence was significantly lower in System 2 than in Systems 1 (z = 3.04, p = 0.0127) and 3 (z = -2.80, p = 0.0261). Mean relative prevalence was 22.30 ± 2.19%, 14.30 ± 1.72%, 21.60 ± 2.17%, and 15.80 ± 2.06% in Systems 1, 2, 3, and 4, respectively (Fig 1 and Table 2). Based on sentinel assays with *T. molitor*, system had a significant effect on the relative prevalence of *M. robertsii* ($\chi^2$ = 11.94, df = 3, p = 0.0076). Similar to assays using *G. mellonella*, relative prevalence was significantly lower in System 2 (z = 3.489, p = 0.0027) than in System 1. Relative prevalence was 34.00 ± 2.43%, 23.50 ± 2.28%, 30.70 ± 2.53%, and 28.20 ± 2.28% in Systems 1, 2, 3, and 4, respectively (Fig 1 and Table 2).

Year had a significant effect on relative prevalence of *M. robertsii* in sentinel assays with *T. molitor* ($\chi^2$ = 14.94, df = 2, p = 0.0005), but not *G. mellonella*. Relative prevalence in 2021 was greater than in 2022 (z = 2.45, p = 0.0377). Relative prevalence was significantly lower in 2022

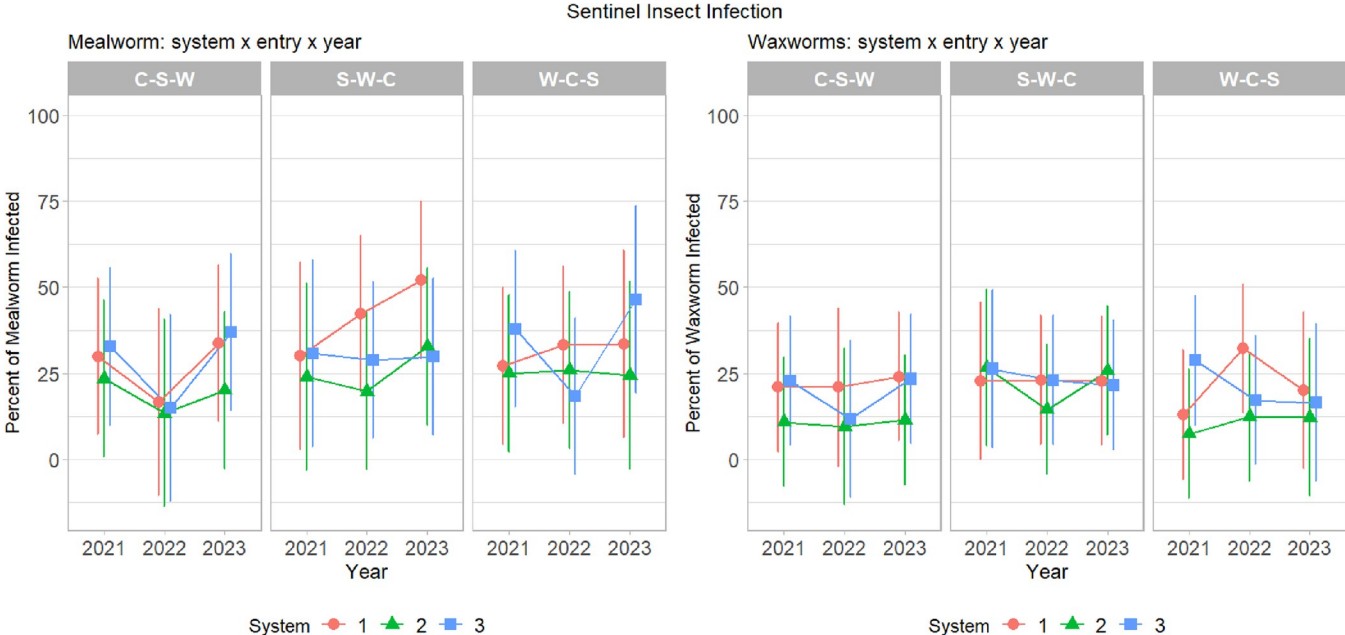

**Fig 1. Proportion of mealworms (*T. molitor*, left panel) and waxworms (*G. mellonella*, right panel) infected by *M. robertsii* by system, entry, and year.** Red circles represent System 1 (inversion tillage), green triangles represent System 2 (high-speed disk), and blue squares represent System 3 (no-till planting). Entries include crop sequences of corn-soybean-wheat (C-S-W), soybean-wheat-corn (S-W-C), and wheat-corn-soybean (W-C-S) in 2021, 2022, and 2023, respectively. Error bars represent the 95% confidence level.

**Table 2. Mean infection rates of G. *mellonella* and *T. molitor* by *M. robertsii* in each cash crop by system.** System had a significant effect for infection rates of *G. mellonella* (P = 0.0056) and *T. molitor* (P = 0.0076). Mean infection rates for the interaction of system and crop were not significantly different.

| Crop | System 1 | System 2 | System 3 | System 4 |
|---|---|---|---|---|
| **% Infection of *G. mellonella*** | | | | |
| Corn | 19.98 ± 3.28 [AB] | 11.65 ± 3.28 [A] | 25.43 ± 3.28 [B] | |
| Soy | 25.03 ± 3.28 | 16.33 ± 3.28 | 20.51 ± 3.28 | |
| Wheat | 21.23± 4.00 | 16.09 ± 4.00 | 18.13 ± 4.00 | |
| Alfalfa | | | | 15.85 ± 2.42 |
| **% Infection of *T. molitor*** | | | | |
| Corn | 34.40 ± 4.73 [A] | 21.58 ± 4.73 [B] | 34.50 ± 4.73 [A] | |
| Soy | 38.40 ± 3.28 [A] | 27.37 ± 3.28 [B] | 27.88 ± 3.28 [B] | |
| Wheat | 27.00 ± 5.46 | 20.55 ± 5.46 | 30.70 ± 5.46 | |
| Alfalfa | | | | 28.16 ± 2.42 |

than 2023 (z = -3.82, p = 0.0040). No other differences were detected between years. Mean relative prevalence of *M. robertsii* based on *T. molitor* infection rate was 28.9 ± 1.90%, 24.60 ± 2.32%, and 33.70 ± 2.18% in 2021, 2022, 2023, respectively (Fig 1 and Table 2). All other pairwise comparisons, including crop and crop within system, were not significant for either insect species.

2) Did relative prevalence of *M. robertsii* based on sentinel *T. molitor* and *G. mellonella* differ?

The estimation of relative prevalence based on mean infection rates of *T. molitor* was greater than estimations of prevalence based on *G. mellonella* ($\chi^2$ = 8.43, df = 1, p = 0.0037). The mean relative prevalence based on *T. molitor* was 29.16 ± 1.26%, whereas relative prevalence based on *G. mellonella* was 18.73 ± 1.04% (Table 2).

1. 3) Did the legacy of differences in tillage used in soybeans in 2021 affect the relative prevalence of *M. robertsii* in corn in 2023?

In corn in 2023, soybean management in Systems 1, 2, or 3 did not have a significant effect on the relative prevalence of *M. robertsii* based on infection rates of *T. molitor* ($\chi^2$ = 3.31, df = 2, p = 0.1909). However, system had a significant effect on the relative prevalence of *M. robertsii* based on infection rates of *G. mellonella* ($\chi^2$ = 14.32, df = 2, p = 0.0008). Relative prevalence in System 1, in which inversion tillage was used in soybean in 2021 and in corn in 2023, was significantly greater than in System 2, which used the HSD in soybean and inversion tillage in corn (z = 3.326, p = 0.0025). Similarly, relative prevalence in System 3, which used no-till planting of soybean and inversion tillage in corn, was greater than in System 2 (z = -3.11, p = 0.0053). Relative prevalence in System 1 was not significantly different than in System 3 (z = 0.211, p = 0.9757). Based on sentinel assays with *G. mellonella*, relative prevalence of *M. robertsii* in corn in 2023 was 24.05 ± 5.79%, 11.46 ± 5.79%, and 23.34 ± 5.79% in Systems 1, 2, and 3, respectively.

1. 4) Did environmental conditions associated with each system affect the relative prevalence of *M. robertsii*?

Principal components analysis explained 12.9% of the adjusted explained variation in disturbance indicators and soil properties among the system-crop combinations (Fig 2). Axis 1 explained 16.61% of the adjusted explained variation and represents level of disturbance and was related to soil organic matter content, cation exchange capacity, and Ca and Mg content. Axis 2 explained 15.84% of the explained variation salts and Zn. The centroids for the corn

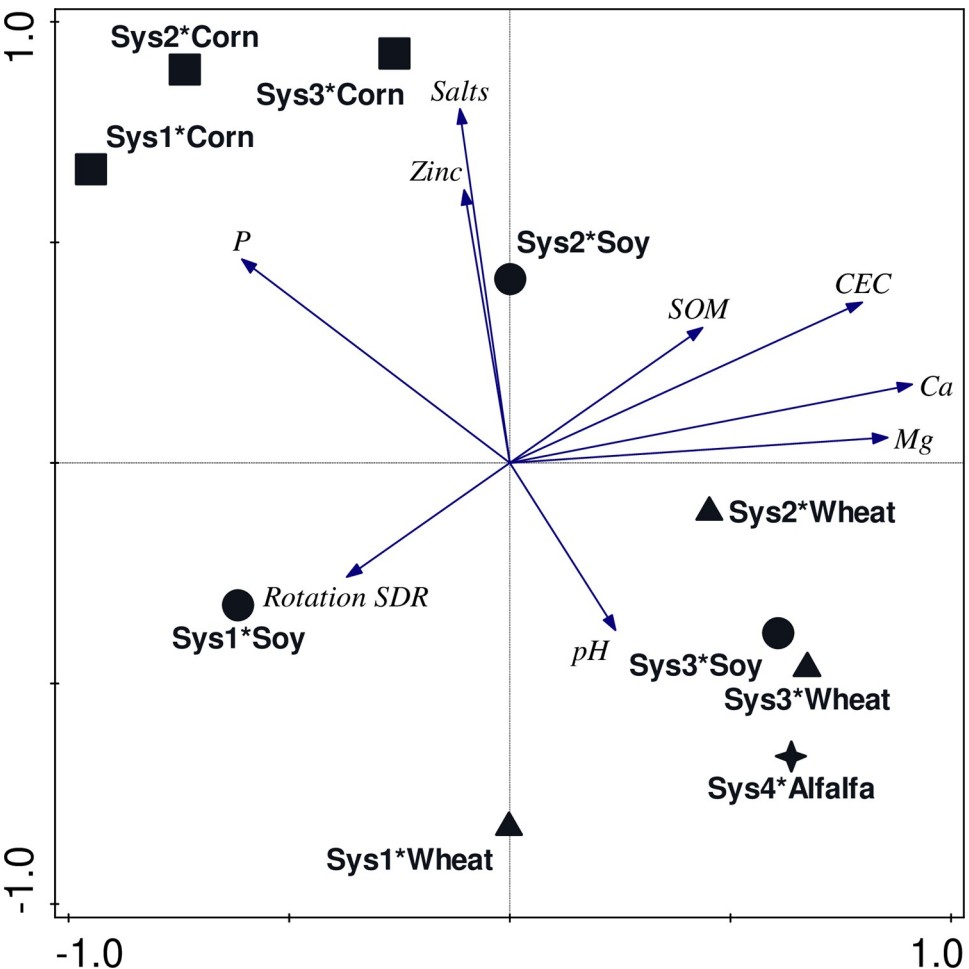

**Fig 2. Biplot from PCA visualizing the relationship between system, crop, and environmental variables.** The interaction of system*crop accounted for 12. 9% of the adjusted variation in disturbance indicators and soil properties. Axes1 and 2 represents 16.61% and 15.84% of the explained variation, respectively. Only environmental variables with at least 20% fit to the ordination space are shown. CEC = Cation exchange capacity, Rotation SDR = Soil disturbance rating accumulated through the three-year rotation, SOM = Soil organic matter (%).

phase in Systems 1, 2, and 3, which were all planted following inversion tillage, clustered in one quadrant of the ordination space and were associated with high levels of disturbance and soil P, Zn, and salts. The centroids for soybean in System 3, managed with no-till planting, and wheat in System 3, which followed no-till planting in the soybean phase, clustered with System 4, perennial alfalfa, which experienced soil disturbance only when planted, and were associated with soil pH and low disturbance. The centroids for the soybean phase in Systems 1 (inversion tillage), 2 (shallow high-speed disk), and 3 (no-till planting) were dispersed across the ordination space according to level of disturbance. The centroid for System 1 soybean, managed with inversion tillage was associated with the soil disturbance rating accumulated across the three-year rotation.

The associations between environmental variables and estimated relative prevalence of *M. robertsii* varied between the two sentinel insect species (Table 3). Across all systems, *G. mellonella* produced lower estimates of relative prevalence of *M. robertsii* than *T. molitor*, but the association with environmental variables was greater for *G. mellonella*. Across all systems, environmental variables explained 28.4% of the variation in estimated prevalence of *M.*

**Table 3. Environmental factors contributing to the variation in relative prevalence *of M. robertsii* in assays with G. *mellonella* and *T. molitor*, determined by of step-wise forward selection multiple regression across all systems and in each of the four cropping systems.**

| | | *G. mellonella* | | | | | *T. molitor* | | |
|---|---|---|---|---|---|---|---|---|---|
| **Variable** | **Estimate** | **St. Error** | **t ratio** | **Pr > \|t\|** | **Variable** | **Estimate** | **St. Error** | **t ratio** | **Pr > \|t\|** |
| **All Systems*** | $r^2_{adj}$ = 0.284, F = 24.76, P < 0.0001 | | | | **All Systems** | $r^2_{adj}$ = 0.048, F = 4.651, P = 0.0004 | | | |
| (Intercept) | -1.972 | 0.266 | -7.40 | <0.0001 | (Intercept) | 1.470 | 0.507 | 2.90 | 0.0040 |
| Soil moist. | -4.139 | 0.454 | -9.12 | <0.0001 | pH | -0.184 | 0.049 | -3.79 | 0.0002 |
| Day of year | 0.002 | 0.000 | 8.76 | <0.0001 | Day of year | -0.001 | 0.000 | -2.69 | 0.0076 |
| S | 0.051 | 0.014 | 3.60 | 0.0004 | Soil moist. | 1.349 | 0.541 | 2.49 | 0.0132 |
| Salts | -0.183 | 0.356 | -3.33 | 0.0010 | Rotation SDR | 0.000 | 0.000 | 2.46 | 0.0146 |
| POX-C | 0.000 | 0.000 | 2.07 | 0.0396 | S | -0.037 | 0.016 | -2.38 | 0.0176 |
| **System 1** | $r^2_{adj}$ = 0.358, F = 11.60, P < 0.0001 | | | | **System 1*** | $r^2_{adj}$ = 0.221, F = 6.39, P < 0.0001 | | | |
| (Intercept) | -0.032 | 0.324 | -0.10 | 0.9207 | (Intercept) | 1.193 | 0.172 | 6.95 | <0.0001 |
| Day of year | 0.002 | 0.000 | 4.84 | <0.0001 | POX-C | -0.002 | 0.000 | -3.26 | 0.0004 |
| Soil moist. | -0.002 | 0.000 | -4.10 | <0.0001 | Annual SDR | 0.004 | 0.001 | 3.10 | 0.0026 |
| CEC | -0.078 | 0.000 | -4.53 | 0.0001 | Annual no. disturbances | -0.059 | 0.022 | -2.75 | 0.0073 |
| POX-C | 0.001 | 0.000 | 2.68 | 0.0087 | Salts | 2.358 | 0.879 | 2.68 | 0.0087 |
| % Clay | 0.016 | 0.007 | 2.36 | 0.0202 | | | | | |
| **System 2** | $r^2_{adj}$ = 0.287, F = 13.77, P < 0.0001 | | | | **System 2** | $r^2_{adj}$ = 0.185, F = 6.40, P = 0.0001 | | | |
| (Intercept) | 0.5826 | 0.154 | 3.76 | 0.0003 | (Intercept) | 0.636 | 0.229 | 2.77 | 0.0069 |
| Day of year | 0.002 | 0.000 | 4.81 | <0.0001 | K | 0.005 | 0.001 | 4.16 | <0.0001 |
| Ca | -0.001 | 0.000 | 13.93 | 0.0002 | POX-C | -0.001 | 0.000 | -3.23 | 0.0017 |
| Cu | -0.056 | 0.027 | -2.09 | 0.0396 | S | -0.058 | 0.025 | -2.29 | 0.0242 |
| | | | | | Annual SDR | 0.001 | 0.000 | 2.24 | 0.0278 |
| **System 3** | $r^2_{adj}$ = 0.106, F = 6.64, P = 0.0020 | | | | **System 3** | $r^2_{adj}$ = 0.135, F = 3.97, P = 0.0026 | | | |
| (Intercept) | 0.081 | 0.097 | 0.83 | 0.4096 | (Intercept) | 0.826 | 0.305 | 2.71 | 0.0080 |
| Day of year | 0.001 | 0.000 | 3.07 | 0.0028 | Ca | -0.001 | 0.000 | -2.69 | 0.0085 |
| Annual SDR | 0.001 | 0.000 | 2.01 | 0.0475 | POX-C | 0.001 | 0.000 | 2.67 | 0.0090 |
| | | | | | S | -0.075 | 0.032 | -2.33 | 0.0220 |
| | | | | | Zn | 0.267 | 0.118 | 2.27 | 0.0257 |
| | | | | | Mg | 0.002 | 0.001 | 2.18 | 0.0320 |
| **System 4** | $r^2_{adj}$ = 0.324, F = 7.43, P < 0.0001 | | | | **System 4** | $r^2_{adj}$ = 0.028, F = 3.92, P = 0.0014 | | | |
| (Intercept) | -1.832 | 0.668 | -2.74 | 0.0080 | (Intercept) | 0.533 | 0.331 | 1.61 | <0.0001 |
| S | 0.095 | 0.030 | 3.14 | 0.0026 | | | | | |
| Soil moist. | -0.000 | 0.000 | -3.08 | 0.0031 | | | | | |
| Days since disturbance | -0.001 | 0.001 | -2.97 | 0.0043 | | | | | |
| Day of year | 0.001 | 0.000 | -2.97 | 0.0113 | | | | | |
| pH | 0.190 | 0.074 | 2.58 | 0.0123 | | | | | |

*All systems, n = 360; Systems 1–3, n = 96; System 4, n = 68. CEC = Cation exchange capacity, Day of year = Julian day of soil assay; EC = Electrical conductivity, POX-C = Permanganate oxidizable carbon, SDR = Soil disturbance rating, Soil moist. = Soil matric potential.

*robertsii* in assays using *G. mellonella*, but only 4.8% of the variation in assays using *T. molitor*. Five environmental variables were significant predictors of the relative prevalence of *M. robertsii* by *G. mellonella* and *T. molitor*. Three predictors—soil moisture, day of year of assay, and soil S content, were common to both sentinel species, although all three were in opposite direction from the other species.

The associations of *M. robertsii* with disturbance indicators and soil properties varied among systems and insect species. In assays using *G. mellonella*, environmental variables

Table 4. Mean soil compaction in soybean in systems 1–3 and alfalfa (System 4).

| | Compaction (mPa)* | | | |
|---|---|---|---|---|
| Depth (cm) | System 1 | System 2 | System 3 | System 4 |
| 2.5 | 1.02 ± 0.18 [A] | 0.89 ± 0.01 [A] | 0.31 ± 0.18 [B] | 0.91 ± 0.11 [A] |
| 5.1 | 1.18 ± 0.20 [A] | 2.01 ± 0.20 [B] | 0.75 ± 0.20 [C] | 1.54 ± 0.09 [B] |
| 7.6 | 1.20 ± 0.22 [A] | 2.27 ± 0.23 [B] | 1.19 ± 0.23 [A] | 1.80 ± 0.13 [B] |
| 10.2 | 1.32 ± 0.24 [A] | 2.24 ± 0.24 [B] | 1.33 ± 0.24 [A] | 1.96 ± 0.12 [B] |

*Different letters in the same row are different at p <0.05.

explained 35.8%, 28.7%, 10.6% and 32.4% of variation in estimated relative prevalence of *M. robertsii* in Systems 1, 2, 3, and 4, respectively (Table 3). The day of year that the soil was collected and assayed was a significant positive predictor in all four systems, indicating that prevalence of *M. robertsii* increased over the growing season. Soil moisture was a significant predictor in Systems 2 (HSD) and 4 (alfalfa), indicating greater prevalence with increases in soil moisture. Other soil properties and disturbance measures associated with relative prevalence of *M. robertsii* but unique to single systems included: a negative predictor, CEC, and two positive predictors, POX-C and percent clay, in System 1; soil Ca and Cu content as negative predictors in System 2; SDR as a positive predictor in System 3; and soil S and pH as positive predictors, and days since disturbance as a negative predictor, in System 4 (Table 3).

In assays using *T. molitor*, environmental variables explained 22.1%, 18.5%, 13.5%, and 2.8% of variation in estimated relative prevalence of *M. robertsii* in Systems 1, 2, 3, and 4, respectively (Table 3). Soil S content was a negative predictor in Systems 2 and 3. The direction of associations, positive or negative, differed among systems for some shared predictors. For example, POX-C was a negative predictor in Systems 1 and 2 and a positive predictor in System 3. Indicators of disturbance were significant predictors in Systems 1 and 2 where annual SDR was a positive predictor, while the number of disturbances was a negative predictor in System 1. No disturbance indicators or soil properties were associated with relative prevalence of *M. robertsii* in System 4.

Soil compaction at 2.5 to 10.2 cm depth differed significantly among systems at 2.5 cm ($F_{3,35.7}$ = 3.66, p = 0.0213), 5.1cm ($F_{3,40}$ = 3.96, p = 0.0145); 7.6cm ($F_{3,36.3}$ = 12.95, p < 0.0001), and 10.2 cm ($F_{3,36.3}$ = 8.48, p = 0.0002) (Table 4). In soybean, the crop phase in in which the main difference in soil management was imposed, compaction in Systems 2 was greater on the two dates measured than in Systems 1 and 3 at 5.1, 7.6, and 10.2 cm depths, but was not different from compaction in System 4 (alfalfa) (Table 4). No measures of compaction were significantly associated with prevalence of *M. robertsii* in assays with either *G. mellonella* or *T. molitor* on the two dates on which compaction was measured.

## Discussion

Tillage is an important practice commonly used by organic growers to incorporate crop and cover crop residues, soil fertility amendments, prepare seedbeds for crops, and manage weeds. Soil disturbance from tillage can contribute to soil erosion, nutrient and organic matter loss, soil health decline, and disruption of naturally occurring insect-parasitic fungal communities [3,14,49]. Organic growers, like non-organic growers, are interested in maintaining or improving the health of their soil [58–61], and one approach to building soil health is to reduce tillage. Means of reducing tillage include reducing the frequency of tillage events, either within a growing season or within a crop rotation or reducing the intensity of disturbance through

choice of tillage implements. Some options for reducing disturbance in organic cropping systems include incorporation of a perennial forage phase in their crop rotation, use of no-till planting into winter cover crops terminated with a roller-crimper, or use of tillage implements such as the shallow high-speed disk, which imposes disturbance to a shallower depth than those typical when using a moldboard plow or chisel plow to reduce disturbance [29,30,35,36,40,62].

## System effect

We quantified the effect of three annual and one perennial cropping systems that varied in intensity and depth of soil disturbance during a three-year crop rotation on the prevalence of entomopathogenic fungus, *M. robertsii*. We hypothesized that tillage frequency and intensity, as estimated by the SDR, and depth of soil disturbance would negatively affect the relative prevalence of *M. robertsii*. We expected that System 1, which used inversion tillage in the soybean and corn phases of the rotation, would result in the lowest relative prevalence of *M. robertsii*. Use of the moldboard plow to a depth of 20 cm in all crop phases was associated with the greatest SDR of the tillage tools utilized. However, we detected greater relative prevalence of *M. robertsii* by both sentinel insect species in System 1, which used a moldboard plow in corn, chisel plow in wheat, and no-till planting in soybean, compared to System 2, which utilized a high-speed disk set at a depth of 5 to 10 cm in soybeans and wheat, and a moldboard plow in corn. We expected that System 3, which utilized no-till planting in the soybean phase, a high-speed disk in wheat, and a moldboard plow in the corn phase, would have a greater relative prevalence of *M. robertsii* than Systems 1 or 2, as System 3 experienced less soil disturbance than Systems 1 or 2. However, because multiple passes were needed to manage plant residues with the high-speed disk, System 2 experienced greater frequency and intensity (SDR) of disturbance than the other three system, although to a shallower depth compared to System 1 (Table 1). Therefore, unexpectedly, the relative prevalence of *M. robertsii* was, on average, lower in System 2 than in Systems 1 and 3, where we observed similar relative prevalences of *M. robertsii*.

In assays with *G. mellonella*, relative prevalence of *M. robertsii* in System 4, which experienced the lowest frequency and intensity of disturbance, was low. Other studies suggest that some level of soil disturbance benefits the detection of *Metarhizium* spp. in agricultural soil. In a survey of *Metarhizium* spp. in soils sampled from a long-term experiment in the Mid-Atlantic region, organically managed plots managed with a chisel plow harbored greater numbers of *Metarhizium* colony-forming units than no-till plots [49]. Another study reported greater prevalence of *Metarhizium* spp. in an organic feed grain rotation in inversion tillage treatments compared to reduced tillage treatments [33]. In that study, the authors suggested that the detection of *Metarhizium* was facilitated by the mixing of spores throughout the soil profile in disturbed systems compared to the reduced tillage systems where fungal spores would remain relatively aggregated around infected insects. Our findings support the hypothesis proposed by Jabbour and Barbercheck [33] in that the reduced tillage system did not provide the soil mixing and distribution of fungal spores throughout the soil profile that the inversion tillage system provided, therefore resulting in lower detection of *M. robertsii* in System 4 than Systems 1 and 3. We hypothesized that System 2 (HSD) would have an intermediate prevalence of *M. robertsii* compared to System 1 (inversion) or 3 (reduced tillage). However, among the three annual cropping systems, we observed the lowest relative prevalence of *M. robertsii* using both *G. mellonella* and *T. molitor* in System 2 (HSD) compared to System 1. We suggest that that some level of soil mixing benefits the detection of *M. robertsii*, while a greater level of disturbance reduces prevalence, especially in systems with relatively low disturbance, such as

Systems 3 (no-till planting in soybean) and 4 (perennial alfalfa). We suggest that the greater frequency and intensity of disturbance associated with System 2 resulted in the observed lower prevalence of *M. robertsii* in that system. Additionally, the shallower disturbance of the soil by the HSD compared with the moldboard plow may have also contributed to the lower detection of *M. robertsii* in System 2, as a greater volume of soil would have been mixed in System 1 compared to System 2 [33].

We expected that the greatest prevalence of *M. robertsii* would occur in System 4 (alfalfa) as it was managed as a perennial forage crop that experienced the least frequent and intense level of disturbance compared to Systems 1, 2, and 3, in which annual crops were grown. However, there were no significant difference in mean relative prevalence associated with crop alone. Another study in the mid-Atlantic US similarly found greater numbers of colony forming units (CFU) of *Metarhizium* spp. under tilled soybean compared with alfalfa [49]. We suggest that *M. robertsii* is well-adapted to the range of levels of disturbance in agricultural systems and can tolerate a wide range of environmental conditions due to its diverse roles as a generalist entomopathogen, saprophyte and endophyte [63–65].

### Legacy effect

We hypothesized that the legacy of the tillage management in soybean in 2021 would affect prevalence of *M. robertsii* in corn in 2023. In System 1, soybeans were managed with inversion tillage in 2021, whereas in System 3, soybeans were managed with no-till planting. This difference in soybean management provided an opportunity to compare the legacy of no-tillage and inversion tillage in soybeans on the corn phase of the rotation. Wheat, the crop following soybeans and preceding corn in 2022, was managed with non-inversion tillage in all systems, including the use of the chisel plow in System 1 and the HSD in Systems 2 and 3. Corn in both Systems 1 and 3 were managed with inversion tillage. We hypothesized we would measure greater relative prevalence of *M. robertsii* in corn after no-till soybean than in corn after soybeans managed with inversion tillage due to the potential negative legacy effects of soil disturbance. However, we did not detect legacy effects of soil disturbance on prevalence of *M. robertsii* in these cropping systems, suggesting that in-season management has a greater effect on the relative prevalence of *M. robertsii* than does management legacy, at least within a single three-year rotation.

### Sentinel insects

We found that relative prevalence based on infection rates of *G. mellonella* by *M. robertsii*, was lower than that for sentinel *T. molitor*. Both sentinel species showed the same general trend in prevalence among systems with lowest relative prevalence in System 2, but no significant differences among crops. In a survey of vineyard and semi-natural habitat soils, Sharma et al. [16] reported that *T. molitor* was more likely than *G. mellonella* to become infected with *M. robertsii* [16]. The authors found that *T. molitor* and *G. mellonella* worked well together to estimate relative prevalence of *M. robertsii* and suggested that using both species as sentinel larvae would provide a greater understanding of the distribution of entomopathogenic fungi than either one alone. Similarly, Vänninen et al. [66] observed high sensitivity of *T. molitor* to *M. anisopliae* in assays of agricultural soils.

### Environmental effects

The four management systems in our study resulted in different soil characteristics and levels of disturbance (Table 1 and Fig 2). Even though our estimates of relative prevalence of *M. robertsii* using *G. mellonella* were lower than in assays using *T. molitor*, the relative prevalence

determined from assays using *G. mellonella* was more closely associated with particular soil properties (Table 3). We identified all *Metarhizium* in this study and in previous studies conducted at this site since 2010 as *M. robertsii* by morphological characteristics and translation elongation factor 1-alpha (TEF 1-α) sequencing [24,25,34,49,67].

We used a commonly employed assay method based on infection of sentinel insects to detect relative prevalence of *M. robertsii*. Using this method, and consistent with our previous results [33,34], we found that *M. robertsii* was the overwhelmingly dominant EPF at our site. Others have used culture-based methods alone or in combination with sentinel insect assays [e.g., 32,49,64,65,68–70] to investigate the prevalence and diversity of EPF in soil and detected greater diversity of EPF than we detected in our study site. It is possible that if we had used additional types of assays, we may have detected a greater diversity of EPF species.

In an assessment of the *Metarhizium* community in soil from a single agricultural field and surrounding hedgerows in Denmark using *T. molitor as a sentinel insect*, Steinwender et al. [64,65] identified multiple genotypes of *M. robertsii* using multilocus simple sequence repeat typing. The authors suggested that multilocus genotypes (MLGs) of *M. robertsii* may have originated from multiple immigration events or that the multiple genotypes represented a long evolutionary history of local lineages at the site. The authors concluded that agricultural practices appear to allow for co-occurrence of MLGs of multiple *Metarhizium* species and that knowledge of the factors responsible for the dominance of particular species or genotypes could be used to either improve conditions for frequently occurring MLGs or to enhance conditions for the addition of rarer MLGs [64,65]. Similarly, Kepler et al [49], using multilocus microsatellite genotyping, found significant genotypic diversity among *M. robertsii* isolates, which fell into multiple MLGs in two clades. Those authors [49] suggested that the multiple levels of diversity could be used to inform strategies by which *Metarhizium* populations in soil could be manipulated to suppress diverse pest organisms and promote plant health. In our study, we did not identify MLGs of *M. robertsii*, so we do not know its genotypic diversity at this site. We speculate that due to its apparently greater susceptibility to *M. robertsii*, *T. molitor* may have "captured" a greater diversity of genotypes of *M. robertsii*, each with different tolerances to specific environmental conditions, than did *G. mellonella*, giving the appearance of non-responsiveness to environmental conditions specific to each crop management system. Our results suggest that the two sentinel insect species provided different but complementary information. *T. molitor* may have provided a better estimate of overall relative prevalence, but *G. mellonella* may have provided more information on the environmental tolerances of the particular genotypes infecting it.

Although we did not detect a significant effect of compaction on relative prevalence of *M. robertsii* on the two dates on which we measured compaction, soil in System 2 was more compacted in the surface 10.2 cm than in Systems 1 and 3 (Table 4). Compaction in the surface 10.2 cm in System 4 was not different from System 2 and mean relative prevalence of *M. robertsii* in assays with *G. mellonella* was low and similar in Systems 2 and 4. The mean compaction in System 2 at 7.6 and 10.2 cm depth exceeded 2.07 mPa, a level of compaction that can inhibit root growth [71]. In a laboratory experiment, Viera Tiago et al. [72] observed a decrease in colony forming units of *Metarhizium* sp. after accidental soil compaction occurred. We suggest that although we did not observe a significant relationship between compaction and relative prevalence of *M. robertsii* for the two corresponding dates on which sentinel assays and measurements of compaction were conducted, more observations and experiments focusing on the effects of soil compaction on relative prevalence are needed to better understand the effects of soil disturbance and compaction on those relationships.

## Conclusion

Organic growers rely on natural processes and cycles, such as biological and microbial control, to prevent damaging populations of insect pests. The relative prevalence of *M. robertsii*, measured as infection rate of sentinel insects, was positively related to soil moisture and progression of the growing season and was not reduced by soil disturbance. Rather, relative prevalence was greater in systems with intermediate levels of soil disturbance. Even though we expected that the high-speed disk would result in an intermediate level of disturbance, it resulted in the highest frequency and intensity of disturbance, greater compaction, and lower relative prevalence of *M. robertsii* than annual cropping systems managed with inversion tillage, chisel plowing, or no-till planting. We suggest that while the high-speed disk may be beneficial or less damaging to soil as an occasional management tool, it had unexpected negative relationships with relative prevalence of *M. robertsii*. To increase prevalence of entomopathogenic fungi such as *M. robertsii*, incorporating diverse soil management practices, such as inversion and non-inversion tillage, and no-till planting, where appropriate and practicable, may be a better solution than the high-speed disk for growers seeking to improve soil health. We also suggest that in-season management has a greater effect than does the legacy of management in the preceding crops. This is promising for growers who are reliant on tillage but concerned about the potential negative impacts of inversion tillage, because the occasional use of inversion tillage will not necessarily have lasting detrimental effects on beneficial soil organisms such as *M. robertsii*.

## Supporting information

**S1 File. S1 Table Three-year crop sequence in full entry cropping systems experiment Systems 1–4. S2 Table.** Management strategies for four organic feed grain and forage systems showing the cropping sequence for Entry 3 (Wheat-Corn-Soybean).S3 Table. Field operations by experimental system, rotation entry point, and date.S4 Table. Number of isolates by experimental system and crop of *Metarhizium* sp. from sentinel T. *molitor* and G. *mellonella* subjected to molecular analysis by sequencing the translation elongation factor-1 alpha (5α-TEF) by the methods of Kepler et al. [49] and described in Ahmad et al. [25] and Randhawa et al. [34]. All isolates were identified as M. *robertsii*.
(DOCX)

## Acknowledgments

We thank I. Ahmad and Z. Velez-Ferrer for molecular identification of *M. robertsii* detected in this experiment. We thank reviewers of early version of the manuscript for their constructive comments to improve the final version.

## Author Contributions

**Conceptualization:** John M. Wallace, Mary E. Barbercheck.

**Data curation:** Shea A. W. Tillotson, Christina A. Voortman, John M. Wallace, Mary E. Barbercheck.

**Formal analysis:** Shea A. W. Tillotson, John M. Wallace.

**Funding acquisition:** John M. Wallace, Mary E. Barbercheck.

**Investigation:** Shea A. W. Tillotson, Christina A. Voortman.

**Methodology:** Shea A. W. Tillotson, Christina A. Voortman.

**Project administration:** Christina A. Voortman, John M. Wallace, Mary E. Barbercheck.

**Validation:** John M. Wallace, Mary E. Barbercheck.

**Visualization:** Shea A. W. Tillotson, Christina A. Voortman, John M. Wallace, Mary E. Barbercheck.

**Writing – original draft:** Shea A. W. Tillotson, Mary E. Barbercheck.

**Writing – review & editing:** Shea A. W. Tillotson, Christina A. Voortman, John M. Wallace, Mary E. Barbercheck.

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
