## [Decision Letter · Decision Letter 0]

8 Oct 2024

PONE-D-24-39841Tillage type and sentinel insect species affect the relative prevalence of the entomopathogenic fungus, Metarhizium robertsii, in soilPLOS ONE

Dear Dr. Barbercheck,

Thank you for submitting your manuscript to PLOS ONE. After careful consideration, we feel that it has merit but does not fully meet PLOS ONE’s publication criteria as it currently stands. Therefore, we invite you to submit a revised version of the manuscript that addresses the points raised during the review process.

**Dear Authors**

**Please refer to reviewers comments especially those related to Metarhizium species identification. Please consider that as a vital piece of your feedback.**

**Good luck**

We look forward to receiving your revised manuscript.

Kind regards,

Rachid Bouharroud

Academic Editor

PLOS ONE

Reviewers' comments:

Reviewer's Responses to Questions

**Comments to the Author**

1. Is the manuscript technically sound, and do the data support the conclusions?

Reviewer #1: Yes

Reviewer #2: Partly

Reviewer #3: Yes

2. Has the statistical analysis been performed appropriately and rigorously? 

Reviewer #1: Yes

Reviewer #2: I Don't Know

Reviewer #3: Yes

3. Have the authors made all data underlying the findings in their manuscript fully available?

Reviewer #1: Yes

Reviewer #2: No

Reviewer #3: Yes

4. Is the manuscript presented in an intelligible fashion and written in standard English?

Reviewer #1: Yes

Reviewer #2: Yes

Reviewer #3: Yes

5. Review Comments to the Author

Reviewer #1: Comments to the Authors

The manuscript titled “Tillage type and sentinel insect species affect the relative prevalence of the entomopathogenic fungus, Metarhizium robertsii, in soil”. This research evaluated the in-season and legacy effects of soil management in four, three-year organic feed grain and forage production systems that varied in number and intensity of soil disturbances on the relative prevalence of the entomopathogenic fungus, Metarhizium robertsii. After carefully reviewing this research I can see some merit in this research but the current form not meet scientific standard so please carefully revise entire manuscript based on the following comments;

1. Line 111: The Tenebrio molitor and Galleria mellonella should be short form. The scientific name of fungi and insect name should be written as short form after first time mentioning carefully fixing these errors throughout the MS.

2. Line 138: check the font.

3. The introduction provides clear details about entomopathogens and their importance, otherwise the reader will be confused.

4. The methodological part should be short and informative, the current form hard to follow.

5. Line 607: which Metarhizium populations? Mention the species.

6. In this manuscript I can see several typographical errors so carefully correct it.

Reviewer #2: The authors of the study performed interesting and long-term research, both in field and laboratory conditions. However, I have doubts about the correct designation of the species Metarhizium robertsii, which was the subject of the research. Doubts concern the lack of molecular tests to confirm species affiliation, which were performed only on the basis of morphological characteristics. Although the authors refer to their previous molecular research and 10 years of experience with the species M. robertsii, in my opinion this is not enough. Especially since, as Bischoff et al. reports in their 2009 study. it is not possible to distinquish taxa M. anisopliae, M. lepidiotae, M. pingshaense and M. robertsii, based on conidial morphology. Therefore, I suggest that the authors re-edit the manuscript and refer the obtained results to Metarhizium spp. I also have a question whether the larvae of both test species were infected only by fungi of the Metarhizium genus, or whether infections caused by e.g. Beauveria genus were also recorded?

Reviewer #3: Dear Authors,

Your manuscript on the effect of soil management on the prevalence of entomopathogenic fungus Metarhizium robertsii is interesting and brings original results with implications for improvement of biodiversity/ecological services of soil in both organic and conventional farming systems. The text of the manuscript is clear, methods are described in enough details, statistical analysis is sound and conclusions are based on obtained data. However, I believe there is some space for improvement as indicated in specific comments below. Numbers mean line numbers in manuscript.

Title: it is fine, describes the study well.

Abstract: I would recommend to add some values documenting the differences between treatments (it is not clear what is magnitude of observed differences, e.g. the range etc.).

Introduction:

I think it would be good to explain why the authors focused only on M. robertsii and did not look at other EPF species.

111 add order and family after species names

113-118 To me it could be reduced just to one hypothesis: "Greater frequency and intensity of soil disturbance has negative impact on relative prevalence of M. robertsii."

118-119 This could be the second hypothesis and authors can support it by citing recent findings on differences between those two bait species in the recoveries of different EPF species (e.g. Beauveria vs. Matarhizium).

243-244 Since these insect species were already mentioned in Introduction, here abbreviated version should be sufficient.

254-257 So there were no other EPF species (e.g. Cordyceps, Beaveria) found on this experimental plot? That is surprising.

494 Tillage has certainly positive role also in pest management of rodents and other pests

496 Perhaps other microogranisms might be mentioned as important, too. E.g. mycoparasitic fungi.

576-586 Some other technique of EPF isolation could be mentioned for comparison. E.g. in some studies EPF were isolated by

culturing soil samples on selective media (Chase et al. 1986 , Correa et al. 2022 , Konopická et al. 2021 , 2022) so it might be useful for readers not familiar with these methods to discuss advantages of live insect-bait technique.

592-594 This species identity might be better placed earlier in Discussion (before discussion on system effect on prevalence of M. robertsii).

632 Better not to limit to insect pests, also mites and other invertebrates are important pets, not mentioning plant pathogens which can also be reduced by ecological services of beneficial microorganisms.

6. PLOS authors have the option to publish the peer review history of their article (what does this mean?). If published, this will include your full peer review and any attached files.

Reviewer #1: **Yes: **Perumal Vivekanandhan

Reviewer #2: No

Reviewer #3: No

---

## [Author Response · Author response to Decision Letter 0]

18 Dec 2024

Response to Reviewers

PONE-D-24-39841

Tillage type and sentinel insect species affect the relative prevalence of the entomopathogenic fungus, Metarhizium robertsii, in soil

We thank the reviewers for their comments and suggestions for improving the manuscript. Please find responses to the comments in BLUE CAPS below.

 Reviewer #1: Comments to the Authors

The manuscript titled “Tillage type and sentinel insect species affect the relative prevalence of the entomopathogenic fungus, Metarhizium robertsii, in soil”. This research evaluated the in-season and legacy effects of soil management in four, three-year organic feed grain and forage production systems that varied in number and intensity of soil disturbances on the relative prevalence of the entomopathogenic fungus, Metarhizium robertsii. After carefully reviewing this research I can see some merit in this research but the current form not meet scientific standard so please carefully revise entire manuscript based on the following comments;

1. Line 111: The Tenebrio molitor and Galleria mellonella should be short form. The scientific name of fungi and insect name should be written as short form after first time mentioning carefully fixing these errors throughout the MS.

REVISED AS SUGGESTED

2. Line 138: check the font.

REVISED AS SUGGESTED

3. The introduction provides clear details about entomopathogens and their importance, otherwise the reader will be confused.

WE BELIEVE THAT PARAGRAPH 2 OF THE INTRODUCTION DESCRIBES THE ROLE OF ENTOMOPATHOGENIC FUNGI AS A PEST SUPPRESSIVE ORGANISMS IN AGROECOSYSTEMS< AND WE CITE IMPORTANT REVIEW PAPERS ON THAT TOPIC. 

4. The methodological part should be short and informative, the current form hard to follow.

WE ACKNOWLEDGE THAT THE METHODS SECTION IS LONG AND DETAILED. THIS WAS A 3-YEAR SYSTEMS EXPERIMENT IN WHICH MANY MANAGEMENT VARIABLES DIFFERED AMONG SYSTEMS. WE HAVE REVISED TO SHORTEN AND IMPROVE CLARITY WHERE POSSIBLE, BUT CONSIDER THE AGRONOMIC DETAIL IMPORTANT TO UNDERSTANDIN THE EXPERIMENT.

5. Line 607: which Metarhizium populations? Mention the species.

IN THE PAPER CITED, KEPLER ET AL. (2015) REFERRED TO MANIPULATION OF THE COMMUNITY OF METARHIZUM SPECIES AT A SITE IN GENERAL, RATHER THAN SPECIFIC SPECIES, HENCE OUR USE OF THE TERM ‘POPULATIONS’.

6. In this manuscript I can see several typographical errors so carefully correct it.

WE HAVE CHECKED THE MS AND CORRECTED TYPOGRAPHICAL ERRORS

Reviewer #2: The authors of the study performed interesting and long-term research, both in field and laboratory conditions. However, I have doubts about the correct designation of the species Metarhizium robertsii, which was the subject of the research. Doubts concern the lack of molecular tests to confirm species affiliation, which were performed only on the basis of morphological characteristics. Although the authors refer to their previous molecular research and 10 years of experience with the species M. robertsii, in my opinion this is not enough. Especially since, as Bischoff et al. reports in their 2009 study. it is not possible to distinquish taxa M. anisopliae, M. lepidiotae, M. pingshaense and M. robertsii, based on conidial morphology. Therefore, I suggest that the authors re-edit the manuscript and refer the obtained results to Metarhizium spp. I also have a question whether the larvae of both test species were infected only by fungi of the Metarhizium genus, or whether infections caused by e.g. Beauveria genus were also recorded?

WE STORED SINGLE SPORE ISOLATES FROM OUR ASSAYS AND HAVE IDENTIFIED A REPRESENTATIVE SAMPLE OF THE ISOLATES BY MOLECULAR METHODS (ADDED TO THE MATERIALS AND METHODS AND SUPPLEMENTARY MATERIAL). ALL ISOLATES WERE IDENTIFIED AS METARHIZIUM ROBERTSII, WHICH IS CONSISTENT WITH OUR RESULTS FROM MORE THAN TEN YEARS OF RESEARCH AT OUR SITE. WE HAVE ISOLATED BEAUVERIA SP. AND ISARIA (CORDYCEPS) SP. EXTREMELY RARELY – ONLY A FEW TIMES DURING THE PAST FIFTEEN YEARS. THEREFORE, DETECTION OF BEAUVERIA WAS EXCLUDED FROM OUR ANALYSES. DURING THIS EXPERMENT< WE DID NOT DETECT ANY OTHER ENTOMOPATHOGENIC FUNGI WITH THE STANDARD, WIDELY PUBLISHED ASSAY METHOD THAT WE USED. THIS IS NOW INDICATED IN THE MS.

Reviewer #3: Dear Authors,

Your manuscript on the effect of soil management on the prevalence of entomopathogenic fungus Metarhizium robertsii is interesting and brings original results with implications for improvement of biodiversity/ecological services of soil in both organic and conventional farming systems. The text of the manuscript is clear, methods are described in enough details, statistical analysis is sound and conclusions are based on obtained data. However, I believe there is some space for improvement as indicated in specific comments below. Numbers mean line numbers in manuscript.

Title: it is fine, describes the study well.

Abstract: I would recommend to add some values documenting the differences between treatments (it is not clear what is magnitude of observed differences, e.g. the range etc.).

EDITED AS SUGGESTED

Introduction:

I think it would be good to explain why the authors focused only on M. robertsii and did not look at other EPF species.

EDITED AS SUGGESTED. M. ROBERTSII IS THE OVERWHELMINGLY DOMINANT SPECIES AT OUR RESEARCH SITE. WE HAVE INDICATED THIS IN THE MATERIALS AND METHODS SECTION UNDER THE SUBHEADING “RELATIVE PREVALENCE OF M. ROBERTSII” AND IN THE DISCUSSION.

111 add order and family after species names

EDITED AS SUGGESTED. WE HAVE ADDED ORDER AND FAMILY NAMES OF TAXA UPON FIRST MENTION IN THE MAIN TEXT OF THE MS.

113-118 To me it could be reduced just to one hypothesis: "Greater frequency and intensity of soil disturbance has negative impact on relative prevalence of M. robertsii."

EDITIED AS SUGGESTED. 

118-119 This could be the second hypothesis and authors can support it by citing recent findings on differences between those two bait species in the recoveries of different EPF species (e.g. Beauveria vs. Matarhizium).

WE KEPT THE HYPOTHESIS ABOUT DIFFERENCES DUE TO SENTINEL SPECIES USED IN ASSAYS AS A THIRD HYPOTHESIS.

243-244 Since these insect species were already mentioned in Introduction, here abbreviated version should be sufficient.

EDITED AS SUGGESTED

254-257 So there were no other EPF species (e.g. Cordyceps, Beaveria) found on this experimental plot? That is surprising.

CORRECT. WE ONLY EXTREMELY RARELY ISOLATE BEUAVERIA AND DID NOT DETECT CORDYCEPS IN THESE EXPERIMETAL PLOTS. THIS FINDING IS CONSISTENT WITH OTHER RESEARCH AT THE SITE SINCE THE SITE WAS FIRST MONITORED IN 2010. WE INDICATE THIS IN THE MATERIALS AND METHODS AND IN THE DISCUSSION.

494 Tillage has certainly positive role also in pest management of rodents and other pests

YES, AGREED

496 Perhaps other microogranisms might be mentioned as important, too. E.g. mycoparasitic fungi.

THAT IS CORRECT, BUT WE DID NOT ISOLATE FUNGI OTHER THAN M. ROBERTSII USING OUR METHODS AND DO NOT WANT TO SPECULATE ON BIOLOGICAL INTERACTIONS THAT MAY HAVE INFLUENCED OUR RESULTS. 

576-586 Some other technique of EPF isolation could be mentioned for comparison. E.g. in some studies EPF were isolated by culturing soil samples on selective media (Chase et al. 1986 , Correa et al. 2022 , Konopická et al. 2021 , 2022) so it might be useful for readers not familiar with these methods to discuss advantages of live insect-bait technique.

EDITED AS SUGGESTED IN DISCUSSION – WE HAVE ADDED TEXT AND CITATIONS FOR KONOPICKÁ 2022 2024; AND NISHI AND SATO 2018 IN THE DISCUSSION. THOSE AUTHORS, IN ADDITION TO CLIFTON ET AL, KEPLER ET AL, and STEINWENDER ET AL (ALREADY CITED IN THE TEXT) USED CULTURE-BASED AND MOLECULAR DETECTION METHODS.

592-594 This species identity might be better placed earlier in Discussion (before discussion on system effect on prevalence of M. robertsii).

EDITED AS SUGGESTED

632 Better not to limit to insect pests, also mites and other invertebrates are important pets, not mentioning plant pathogens which can also be reduced by ecological services of beneficial microorganisms.

EDITED AS SUGGESTED. EDITED TO REFLECT PESTS IN GENERAL THAT WOULD BE AFFECTED BY METARHIZIUM AS AN ENTOMOPATHOGEN AND AN DENDOPHYTE (ARTHROPODS AND PHYTOPATHOGENS)

---

## [Editor Report · Decision Letter 1]

22 Dec 2024

Tillage type and sentinel insect species affect the relative prevalence of the entomopathogenic fungus, Metarhizium robertsii, in soil

PONE-D-24-39841R1

Dear Dr. Barbercheck,

We’re pleased to inform you that your manuscript has been judged scientifically suitable for publication and will be formally accepted for publication once it meets all outstanding technical requirements.

Kind regards,

Rachid Bouharroud

Academic Editor

PLOS ONE
---

## [Editor Report · Acceptance letter]

2 Jan 2025

PONE-D-24-39841R1 

PLOS ONE

Dear Dr. Barbercheck, 

I'm pleased to inform you that your manuscript has been deemed suitable for publication in PLOS ONE. Congratulations! Your manuscript is now being handed over to our production team.

Kind regards, 

on behalf of

Dr. Rachid Bouharroud 

Academic Editor

PLOS ONE